# On two ways to use determinantal point processes for Monte Carlo integration

**Guillaume Gautier**[†*]
g.gautier@inria.fr

**Rémi Bardenet**[†]
remi.bardenet@gmail.com

**Michal Valko**[‡*†]
valkom@deepmind.com

[†]Univ. Lille, CNRS, Centrale Lille, UMR 9189 – CRIStAL, 59651 Villeneuve d'Ascq, France
[*]Inria Lille-Nord Europe, 40 avenue Halley 59650 Villeneuve d'Ascq, France
[‡]DeepMind Paris, 14 Rue de Londres, 75009 Paris, France

## Abstract

When approximating an integral by a weighted sum of function evaluations, determinantal point processes (DPPs) provide a way to enforce repulsion between the evaluation points. This negative dependence is encoded by a kernel. Fifteen years before the discovery of DPPs, Ermakov & Zolotukhin (EZ, 1960) had the intuition of sampling a DPP and solving a linear system to compute an unbiased Monte Carlo estimator of the integral. In the absence of DPP machinery to derive an efficient sampler and analyze their estimator, the idea of Monte Carlo integration with DPPs was stored in the cellar of numerical integration. Recently, Bardenet & Hardy (BH, 2019) came up with a more natural estimator with a fast central limit theorem (CLT). In this paper, we first take the EZ estimator out of the cellar, and analyze it using modern arguments. Second, we provide an efficient implementation[1] to sample exactly a particular multidimensional DPP called *multivariate Jacobi ensemble*. The latter satisfies the assumptions of the aforementioned CLT. Third, our new implementation lets us investigate the behavior of the two unbiased Monte Carlo estimators in yet unexplored regimes. We demonstrate experimentally good properties when the kernel is adapted to basis of functions in which the integrand is sparse or has fast-decaying coefficients. If such a basis and the level of sparsity are known (e.g., we integrate a linear combination of kernel eigenfunctions), the EZ estimator can be the right choice, but otherwise it can display an erratic behavior.

## 1  Introduction

Numerical integration is a core task of many machine learning applications, including most Bayesian methods (Robert, 2007). Both deterministic (Davis & Rabinowitz, 1984; Dick & Pillichshammer, 2010) and random (Robert & Casella, 2004) algorithms have been proposed; see also (Evans & Swartz, 2000) for a survey. All methods require evaluating the integrand at carefully chosen points, called *quadrature nodes*, and combining these evaluations to minimize the approximation error.

Recently, a stream of work has made use of prior knowledge on the smoothness of the integrand using kernels. Oates et al. (2017) and Liu & Lee (2017) used kernel-based control variates, splitting the computational budget into regressing the integrand and integrating the residual. Bach (2017) looked for the best way to sample i.i.d. nodes and combine the resulting evaluations. Finally, Bayesian quadrature (O'Hagan, 1991; Huszár & Duvenaud, 2012; Briol et al., 2015), herding (Chen et al., 2010; Bach et al., 2012), or the biased importance sampling estimate of Delyon & Portier (2016) all favor *dissimilar* nodes, where dissimilarity is measured by a kernel. Our work falls in this last cluster.

We build on the particular approach of Bardenet & Hardy (2019) for Monte Carlo integration based on projection *determinantal point processes* (DPPs, Hough et al., 2006; Kulesza & Taskar, 2012). DPPs are a repulsive distribution over configurations of points, where repulsion is again parametrized by a kernel. In a sense, DPPs are the kernel machines of point processes.

Fifteen years before Macchi (1975) even formalized DPPs, Ermakov & Zolotukhin (EZ, 1960) had the intuition to use a determinantal structure to sample quadrature nodes, followed by solving a linear system to aggregate the evaluations of the integrand into an unbiased estimator. This linear system yields a simple and interpretable characterization of the variance of their estimator. Ermakov & Zolotukhin's result did not diffuse much[2] in the Monte Carlo community, partly because the mathematical and computational machinery to analyze and implement it was not available. Seemingly unaware of this previous work, Bardenet & Hardy (2019) came up with a more natural estimator of the integral of interest, and they could build upon the thorough study of DPPs in random matrix theory (Johansson, 2006) to obtain a fast central limit theorem (CLT). Since then, DPPs with stationary kernels have also been used by Mazoyer et al. (2019) for Monte Carlo integration. In any case, these DPP-based Monte Carlo estimators crucially depend on efficient sampling procedures for the corresponding (potentially multidimensional) DPP.

**Our contributions.** First, we reveal the close link between DPPs and the approach of Ermakov & Zolotukhin (1960). Second, we provide a simple proof of their result and survey the properties of the EZ estimator with modern arguments. In particular, when the integrand is a linear combination of the eigenfunctions of the kernel of the underlying DPP, the corresponding Fourier-like coefficients can be estimated with zero variance. In other words, one sample of the corresponding DPP yields perfect interpolation of the underlying integrand, by solving a linear system. Third, we propose an efficient Python implementation for exact sampling of a particular DPP, called *multivariate Jacobi ensemble*. The code[1] is available in the DPPy toolbox of Gautier et al. (2019). This implementation allows to numerically investigate the behavior of the two Monte Carlo estimators derived by Bardenet & Hardy (2019) and Ermakov & Zolotukhin (1960), in regimes yet unexplored for any of the two. Fourth, important theoretical properties of both estimators, like the CLT of (Bardenet & Hardy, 2019), are technically involved. A CLT for EZ promises to be even more difficult to establish. The current empirical investigation provides a motivation and guidelines for more theoretical work. Our point is not to compare DPP-based Monte Carlo estimators to the wide choice of numerical integration algorithms, but to get a fine understanding of their properties so as to fine-tune their design and guide theoretical developments.

## 2 Quadrature, DPPs, and the multivariate Jacobi ensemble

In this section, we quickly survey classical quadrature rules. Then, we define DPPs and give the key properties that make them useful for Monte Carlo integration. Finally, among so-called *projection* DPPs, we introduce the multivariate Jacobi ensemble used by Bardenet & Hardy (2019) to generate quadrature nodes, and on which we base our experimental work.

### 2.1 Standard quadrature

Following Briol et al. (2015, Section 2.1), let $\mu(\mathrm{d}x) = \omega(x)\,\mathrm{d}x$ be a positive Borel measure on $\mathbb{X} \subset \mathbb{R}^d$ with finite mass and density $\omega$ w.r.t. the Lebesgue measure. This paper aims to compute integrals of the form $\int f(x)\mu(\mathrm{d}x)$ for some test function $f : \mathbb{X} \to \mathbb{R}$. A quadrature rule approximates such integrals as a weighted sum of evaluations of $f$ at some *nodes* $\{x_1, \ldots, x_N\} \subset \mathbb{X}$,

$$\int f(x)\mu(\mathrm{d}x) \approx \sum_{n=1}^{N} \omega_n f(x_n), \tag{1}$$

where the weights $\omega_n \triangleq \omega_n(x_1, \ldots, x_N)$ do not need to be non-negative nor sum to one.

Among the many quadrature designs mentioned in the introduction (Evans & Swartz, 2000, Section 5), we pay special attention to the textbook example of the (deterministic) Gauss-Jacobi rule. This scheme applies to dimension $d = 1$, for $\mathbb{X} \triangleq [-1, 1]$ and $\omega(x) \triangleq (1 - x)^a(1 + x)^b$ with $a, b > -1$. In this case, the nodes $\{x_1, \ldots, x_N\}$ are taken to be the zeros of $p_N$, the orthonormal Jacobi polynomial of degree $N$, and the weights $\omega_n \triangleq 1/K(x_n, x_n)$ with $K(x, x) \triangleq \sum_{k=0}^{N-1} p_k(x)^2$. In particular, this specific quadrature rule allows to perfectly integrate polynomials up to degree $2N - 1$ (Davis & Rabinowitz, 1984, Section 2.7). In a sense, the DPPs of Bardenet & Hardy (2019) are a random, multivariate generalization of Gauss-Jacobi quadrature, as we shall see in Section 3.1.

Monte Carlo integration can be defined as random choices of nodes in (1). Importance sampling, for instance, corresponds to i.i.d. nodes, while Markov chain Monte Carlo corresponds to nodes drawn from a carefully chosen Markov chain; see, e.g., Robert & Casella (2004) for more details. Finally, quasi-Monte Carlo (QMC, Dick & Pillichshammer, 2010) applies to $\mu$ uniform over a compact subset of $\mathbb{R}^d$, and constructs deterministic nodes that spread uniformly, as measured by their *discrepancy*.

## 2.2 Projection DPPs

DPPs can be understood as a parametric class of point processes, specified by a base measure $\mu$ and a kernel $K : \mathbb{X} \times \mathbb{X} \to \mathbb{C}$. The latter is commonly assumed to be Hermitian and trace-class. For the resulting process to be well defined, it is necessary and sufficient that the kernel $K$ is positive semi-definite with eigenvalues in $[0, 1]$, see, e.g., Soshnikov (2000, Theorem 3). When the eigenvalues further belong to $\{0, 1\}$, we speak of a *projection* kernel and a *projection* DPP. One practical feature of projection DPPs is that they almost surely produce samples with fixed cardinality, equal to the rank $N$ of the kernel. More generally, they are the building blocks of DPPs. Indeed, under general assumptions, all DPPs are mixtures of projection DPPs (Hough et al., 2006, Theorem 7). Hereafter, unless specifically stated, $K$ is assumed to be a real-valued, symmetric, projection kernel.

One way to define a projection DPP with $N$ points is to take $N$ functions $\phi_0, \dots, \phi_{N-1}$ orthonormal w.r.t. $\mu$, i.e., $\langle \phi_k, \phi_\ell \rangle \triangleq \int \phi_k(x)\phi_\ell(x)\mu(\mathrm{d}x) = \delta_{k\ell}$, and consider the kernel $K_N$ associated to the orthogonal projector onto $\mathcal{H}_N \triangleq \mathrm{span}\{\phi_k, \ 0 \le k \le N - 1\}$, i.e.,

$$K_N(x, y) \triangleq \sum_{k=0}^{N-1} \phi_k(x)\phi_k(y). \tag{2}$$

We say that the set $\{\mathbf{x}_1, \dots, \mathbf{x}_N\} \subset \mathbb{X}$ is drawn from the projection DPP with base measure $\mu$ and kernel $K_N$, denoted by $\{\mathbf{x}_1, \dots, \mathbf{x}_N\} \sim \mathrm{DPP}(\mu, K_N)$, when $(\mathbf{x}_1, \dots, \mathbf{x}_N)$ has joint distribution

$$\frac{1}{N!} \det(K_N(x_p, x_n))_{p,n=1}^N \mu^{\otimes N}(\mathrm{d}x). \tag{3}$$

$\mathrm{DPP}(\mu, K_N)$ indeed defines a probability measure over sets since (3) is invariant by permutation and the orthonormality of the $\phi_k$s yields the normalization. See also Appendix A.1 for more details on the construction of projection DPPs from sets of linearly independent functions.

The repulsion of projection DPPs may be understood geometrically by considering the Gram formulation of the kernel (2), namely

$$K_N(x, y) = \Phi(x)^\top \Phi(y), \quad \text{where} \quad \Phi(x) \triangleq (\phi_0(x), \dots, \phi_{N-1}(x))^\top. \tag{4}$$

This allows to rewrite the joint distribution (3) as

$$\frac{1}{N!} \underbrace{\det \mathbf{\Phi}(x_{1:N})\mathbf{\Phi}(x_{1:N})^\top}_{=(\det \mathbf{\Phi}(x_{1:N}))^2} \mu^{\otimes N}(\mathrm{d}x), \quad \text{where} \quad \mathbf{\Phi}(x_{1:N}) \triangleq \begin{pmatrix} \phi_0(x_1) & \dots & \phi_{N-1}(x_1) \\ \vdots & & \vdots \\ \phi_0(x_N) & \dots & \phi_{N-1}(x_N) \end{pmatrix}. \tag{5}$$

Thus, the larger the determinant of the *feature matrix* $\mathbf{\Phi}(x_{1:N})$, i.e., the larger the volume of the parallelotope spanned by the *feature vectors* $\Phi(x_1), \dots, \Phi(x_N)$, the more likely $x_1, \dots, x_N$ co-occur.

## 2.3 The multivariate Jacobi ensemble

In this part, we specify a projection kernel. We follow Bardenet & Hardy (2019) and take its eigenfunctions to be multivariate orthonormal polynomials. In dimension $d = 1$, letting $(\phi_k)_{k \ge 0}$ in (2) be the orthonormal polynomials w.r.t. $\mu$ results in a projection DPP called an *orthogonal polynomial ensemble* (OPE, König, 2004). When $d > 1$, orthonormal polynomials can still be uniquely defined by applying the Gram-Schmidt procedure to a set of monomials, provided the base measure is not pathological. However, there is no natural order on multivariate monomials: an ordering $\mathfrak{b} : \mathbb{N}^d \to \mathbb{N}$ must be picked before we apply Gram-Schmidt to the monomials in $L^2(\mu)$. We follow Bardenet & Hardy (2019, Section 2.1.3) and consider multi-indices $k \triangleq (k^1, \dots, k^d) \in \mathbb{N}^d$ ordered by their maximum degree $\max_i k^i$, and for constant maximum degree, by the usual lexicographic order. We still denote the corresponding multivariate orthonormal polynomials by $(\phi_k)_{k \in \mathbb{N}^d}$.

By multivariate OPE we mean the projection DPP with base measure $\mu(\mathrm{d}x) \triangleq \omega(x)\,\mathrm{d}x$ and orthogonal projection kernel $K_N(x,y) \triangleq \sum_{\mathfrak{b}(k)=0}^{N-1} \phi_k(x)\phi_k(y)$. When the base measure is separable, i.e., $\omega(x) = \omega^1(x^1) \times \cdots \times \omega^d(x^d)$, multivariate orthonormal polynomials are products of univariate ones, and the kernel (2) reads

$$K_N(x,y) = \sum_{\mathfrak{b}(k)=0}^{N-1} \prod_{i=1}^{d} \phi_{k^i}^i(x^i)\phi_{k^i}^i(y^i), \tag{6}$$

where $(\phi_\ell^i)_{\ell \geq 0}$ are the orthonormal polynomials w.r.t. $\omega^i(z)\,\mathrm{d}z$. For $\mathbb{X} = [-1,1]^d$ and $\omega^i(z) = (1-z)^{a^i}(1+z)^{b^i}$, with $a^i, b^i > -1$, the resulting DPP is called a *multivariate Jacobi ensemble*.

## 3 Monte Carlo integration with projection DPPs

Our goal is to design random quadrature rules (1) on $\mathbb{X} \triangleq [-1,1]^d$ with desirable properties. We focus on computing $\int f(x)\mu(\mathrm{d}x)$ with the two unbiased DPP-based Monte Carlo estimators of Bardenet & Hardy (BH, 2019) and Ermakov & Zolotukhin (EZ, 1960). We start by presenting the natural BH estimator which, when associated to the multivariate Jacobi ensemble, comes with a CLT with a faster rate than classical Monte Carlo. Then, we survey the properties of the less obvious EZ estimator. Using a generalization of the Cauchy-Binet formula we provide a slight improvement of the key result of EZ. Despite the lack of result illustrating a fast convergence rate, the EZ estimator has a practical and interpretable variance. In particular, this estimator turns a single DPP sample into a perfect integrator as well as a perfect interpolator of functions that are linear combinations of eigenfunctions of the associated kernel. Finally, we detail our exact sampling procedure for multivariate Jacobi ensemble, which allows to exploit the best of both the BH and EZ estimators.

### 3.1 A natural estimator

For $f \in L^1(\mu)$, Bardenet & Hardy (2019) consider

$$\widehat{I}_N^{\mathrm{BH}}(f) \triangleq \sum_{n=1}^{N} \frac{f(\mathbf{x}_n)}{K_N(\mathbf{x}_n, \mathbf{x}_n)}, \tag{7}$$

as an unbiased estimator of $\int f(x)\mu(\mathrm{d}x)$, with variance (see, e.g., Lavancier et al., 2012, Section 2.1)

$$\mathbb{V}\mathrm{ar}\left[\widehat{I}_N^{\mathrm{BH}}(f)\right] = \frac{1}{2} \int \left( \frac{f(x)}{K_N(x,x)} - \frac{f(y)}{K_N(y,y)} \right)^2 K_N(x,y)^2 \mu(\mathrm{d}x)\mu(\mathrm{d}y), \tag{8}$$

which clearly captures a notion of smoothness of $f$ w.r.t. $K_N$ but its interpretation is not obvious.

For $\mathbb{X} = [-1,1]^d$, the interest in multivariate Jacobi ensemble among DPPs comes from the fact that (7) can be understood as a (randomized) multivariate counterpart of the Gauss-Jacobi quadrature introduced in Section 2.1. Moreover, for $f$ essentially $\mathcal{C}^1$, Bardenet & Hardy (2019, Theorem 2.7) proved a CLT with faster-than-classical-Monte-Carlo decay,

$$\sqrt{N^{1+1/d}}\left( \widehat{I}_N^{\mathrm{BH}}(f) - \int f(x)\mu(\mathrm{d}x) \right) \xrightarrow[N \to \infty]{\mathrm{law}} \mathcal{N}\big(0, \Omega_{f,\omega}^2\big), \tag{9}$$

with $\Omega_{f,\omega}^2 \triangleq \frac{1}{2}\sum_{k \in \mathbb{N}^d}(k^1 + \cdots + k^d)\mathcal{F}_{\frac{f\omega}{\omega_{\mathrm{eq}}}}(k)^2$, where $\mathcal{F}_g$ denotes the Fourier transform of $g$, and $\omega_{\mathrm{eq}}(x) \triangleq 1/\prod_{i=1}^{d} \pi\sqrt{1-(x^i)^2}$. In the fast CLT (9), the asymptotic variance is governed by the smoothness of $f$ since $\Omega_{f,\omega}$ is a measure of the decay of the Fourier coefficients of the integrand.

### 3.2 The Ermakov-Zolotukhin estimator

We start by stating the main finding of Ermakov & Zolotukhin (1960), see also Evans & Swartz (2000, Section 6.4.3) and references therein. To the best of our knowledge, we are the first to make the connection with projection DPPs, as defined in Section 2.2. This allows us to give a slight improvement and provide a simpler proof of the original result, based on a generalization of the Cauchy-Binet formula (Johansson, 2006). Finally, we apply Ermakov & Zolotukhin's (1960) technique to build an unbiased estimator of $\int f(x)\mu(\mathrm{d}x)$, which comes with a practical and interpretable variance.

**Theorem 1.** *Consider $f \in L^2(\mu)$ and $N$ functions $\phi_0, \ldots, \phi_{N-1} \in L^2(\mu)$ orthonormal w.r.t. $\mu$. Let $\{\mathbf{x}_1, \ldots, \mathbf{x}_N\} \sim \mathrm{DPP}(\mu, K_N)$, with $K_N(x, y) = \sum_{k=0}^{N-1} \phi_k(x)\phi_k(y)$. Consider the linear system*

$$\begin{pmatrix} \phi_0(\mathbf{x}_1) & \ldots & \phi_{N-1}(\mathbf{x}_1) \\ \vdots & & \vdots \\ \phi_0(\mathbf{x}_N) & \ldots & \phi_{N-1}(\mathbf{x}_N) \end{pmatrix} \begin{pmatrix} y_1 \\ \vdots \\ y_N \end{pmatrix} = \begin{pmatrix} f(\mathbf{x}_1) \\ \vdots \\ f(\mathbf{x}_N) \end{pmatrix}. \tag{10}$$

*Then, the solution of (10) is unique, $\mu$-almost surely, with coordinates $y_k = \frac{\det \mathbf{\Phi}_{\phi_{k-1}, f}(\mathbf{x}_{1:N})}{\det \mathbf{\Phi}(\mathbf{x}_{1:N})}$, where $\mathbf{\Phi}_{\phi_{k-1}, f}(\mathbf{x}_{1:N})$ is the matrix obtained by replacing the $k$-th column of $\mathbf{\Phi}(\mathbf{x}_{1:N})$ by $f(\mathbf{x}_{1:N})$. Moreover, for all $1 \leq k \leq N$, the coordinate $y_k$ of the solution vector satisfies*

$$\mathbb{E}[y_k] = \langle f, \phi_{k-1} \rangle, \quad \text{and} \quad \mathbb{V}\mathrm{ar}[y_k] = \|f\|^2 - \sum_{\ell=0}^{N-1} \langle f, \phi_\ell \rangle^2. \tag{11}$$

*We improved the original result by showing that $\mathbb{C}\mathrm{ov}[y_j, y_k] = 0$, for all $1 \leq j \neq k \leq N$.*

Before we provide the proof, also detailed in Appendix A.2, several remarks are in order. We start by considering a function $f \triangleq \sum_{k=0}^{M-1} \langle f, \phi_k \rangle \phi_k$, $1 \leq M \leq \infty$, where $(\phi_k)_{k \geq 0}$ forms an orthonormal basis of $L^2(\mu)$, e.g., the Fourier basis or wavelet bases (Mallat & Peyré, 2009). Next, we build the orthogonal projection kernel $K_N$ onto $\mathcal{H}_N \triangleq \mathrm{span}\{\phi_0, \ldots, \phi_{N-1}\}$ as in (2). Then, Theorem 1 shows that solving (10), with points $\{\mathbf{x}_1, \ldots, \mathbf{x}_N\} \sim \mathrm{DPP}(\mu, K_N)$, provides unbiased estimates of the $N$ Fourier-like coefficients $(\langle f, \phi_k \rangle)_{k=0}^{N-1}$. Remarkably, these estimates are uncorrelated and have the same variance (11) equal to the residual $\sum_{k=N}^{\infty} \langle f, \phi_k \rangle^2$. The faster the decay of the coefficients, the smaller the variance. In particular, for $M \leq N$, i.e., $f \in \mathcal{H}_N$, the estimators have zero variance. Put differently, $f$ can be reconstructed perfectly from only one sample of $\mathrm{DPP}(\mu, K_N)$.

*Proof.* First, the joint distribution (5) of $(\mathbf{x}_1, \ldots, \mathbf{x}_N)$ is proportional to $(\det \mathbf{\Phi}(x_{1:N}))^2 \mu^{\otimes N}(x)$. Thus, the matrix $\mathbf{\Phi}(\mathbf{x}_{1:N})$ defining the linear system (10) is invertible, $\mu$-almost surely, and the expression of the coordinates follows from Cramer's rule. Then, we treat the case $k = 1$, the others follow the same lines. The proof relies on the orthonormality of the $\phi_k$s and a generalization of the Cauchy-Binet formula (A.1), cf. Lemma A. The expectation in (11) reads

$$\mathbb{E}\left[ \frac{\det \mathbf{\Phi}_{\phi_0, f}(\mathbf{x}_{1:N})}{\det \mathbf{\Phi}(\mathbf{x}_{1:N})} \right] \stackrel{(5)}{=} \frac{1}{N!} \int \det \mathbf{\Phi}_{\phi_0, f}(x_{1:N}) \det \mathbf{\Phi}(x_{1:N}) \, \mu^{\otimes N}(\mathrm{d}x)$$

$$\stackrel{(A.1)}{=} \det \begin{pmatrix} \langle f, \phi_0 \rangle & (\langle f, \phi_\ell \rangle)_{\ell=1}^{N-1} \\ 0_{N-1,1} & I_{N-1} \end{pmatrix} = \langle f, \phi_0 \rangle. \tag{12}$$

Similarly, the second moment reads

$$\mathbb{E}\left[ \left( \frac{\det \mathbf{\Phi}_{\phi_0, f}(\mathbf{x}_{1:N})}{\det \mathbf{\Phi}(\mathbf{x}_{1:N})} \right)^2 \right] \stackrel{(5)}{=} \frac{1}{N!} \int \det \mathbf{\Phi}_{\phi_0, f}(x_{1:N}) \det \mathbf{\Phi}_{\phi_0, f}(x_{1:N}) \, \mu^{\otimes N}(\mathrm{d}x)$$

$$\stackrel{(A.1)}{=} \det \begin{pmatrix} \|f\|^2 & (\langle f, \phi_\ell \rangle)_{\ell=1}^{N-1} \\ (\langle f, \phi_k \rangle)_{k=1}^{N-1} & I_{N-1} \end{pmatrix} = \|f\|^2 - \sum_{k=1}^{N-1} \langle f, \phi_k \rangle^2. \tag{13}$$

Finally, the variance in (11) = (13) - (12)$^2$. The covariance is treated in Appendix A.2. $\qquad\square$

In the setting of the multivariate Jacobi ensemble described in Section 2.3, the first orthonormal polynomial $\phi_0$ is constant, equal to $\mu([-1, 1]^d)^{-1/2}$. Hence, a direct application of Theorem 1 yields

$$\widehat{I}_N^{\mathrm{EZ}}(f) \triangleq \frac{y_1}{\phi_0} = \mu([-1, 1]^d)^{1/2} \frac{\det \mathbf{\Phi}_{\phi_0, f}(\mathbf{x}_{1:N})}{\det \mathbf{\Phi}(\mathbf{x}_{1:N})}, \tag{14}$$

as an unbiased estimator of $\int_{[-1,1]^d} f(x)\mu(\mathrm{d}x)$, see Appendix A.3. We also show that (14) can be viewed as a quadrature rule (1) with weights summing to $\mu([-1, 1]^d)$. Unlike the variance of $\widehat{I}_N^{\mathrm{BH}}(f)$ in (8), the variance of $\widehat{I}_N^{\mathrm{EZ}}(f)$ clearly reflects the accuracy of the approximation of $f$ by its projection onto $\mathcal{H}_N$. In particular, it allows to integrate and interpolate polynomials up to "degree" $\flat^{-1}(N-1)$, perfectly. Nonetheless, its limiting theoretical properties, like a CLT, look hard to establish. In particular, the dependence of each quadrature weight on all quadrature nodes makes the estimator a peculiar object that doesn't fit the assumptions of traditional CLTs for DPPs (Soshnikov, 2000).

### 3.3 How to sample from projection DPPs and the multivariate Jacobi ensemble

To perform Monte Carlo integration with DPPs, it is crucial to sample the points and evaluate the weights efficiently. However, sampling from continuous DPPs remains a challenge. In this part, we review briefly the main technique for projection DPP sampling before we develop our method to generate samples from the multivariate Jacobi ensemble. The code[1] is available in the DPPy toolbox of Gautier et al. (2019), the associated documentation[3] contains a lot more details on DPP sampling.

In both finite and continuous cases, except for some specific instances, exact DPP sampling usually requires the spectral decomposition of the underlying kernel (Lavancier et al., 2012, Section 2.4). However, for projection DPPs, prior knowledge of the eigenfunctions is not necessary, only an oracle to evaluate the kernel is required. Next, we describe the generic exact sampler of Hough et al. (2006, Algorithm 18). It is based on the chain rule and valid exclusively for projection DPPs.

For simplicity, consider a projection $\mathrm{DPP}(\mu, K_N)$ with $\mu(\mathrm{d}x) = \omega(x)\,\mathrm{d}x$ and $K_N$ as in (2). This DPP has exactly $N$ points, $\mu$-almost surely (Hough et al., 2006, Lemma 17). To get a valid sample $\{\mathbf{x}_1, \ldots, \mathbf{x}_N\}$, it is enough to apply the chain rule to sample $(\mathbf{x}_1, \ldots, \mathbf{x}_N)$ and forget the order the points were selected. The chain rule scheme can be derived from two different perspectives. Either using Schur complements to express the determinant in the joint distribution (3),

$$\frac{K_N(x_1, x_1)}{N}\omega(x_1)\,\mathrm{d}x_1 \prod_{n=2}^{N} \frac{K_N(x_n, x_n) - \mathbf{K}_{n-1}(x_n)^{\mathsf{T}}\mathbf{K}_{n-1}^{-1}\mathbf{K}_{n-1}(x_n)}{N - (n-1)}\omega(x_n)\,\mathrm{d}x_n, \qquad (15)$$

where $\mathbf{K}_{n-1}(\cdot) = (K_N(x_1, \cdot), \ldots, K_N(x_{n-1}, \cdot))^{\mathsf{T}}$, and $\mathbf{K}_{n-1} = (K_N(x_p, x_q))_{p,q=1}^{n-1}$. Or geometrically using the base×height formula to express the squared volume in the joint distribution (5),

$$\frac{\|\Phi(x_1)\|^2}{N}\omega(x_1)\,\mathrm{d}x_1 \prod_{n=2}^{N} \frac{\mathrm{distance}^2\big(\Phi(x_n), \mathrm{span}\{\Phi(x_p)\}_{p=1}^{n-1}\big)}{N - (n-1)}\omega(x_n)\,\mathrm{d}x_n. \qquad (16)$$

Note that the numerators in (15) correspond to the incremental posterior variances of a noise-free Gaussian process model with kernel $K_N$ (Rasmussen & Williams, 2006), giving yet another intuition for repulsion. When using the chain rule, the practical challenge is twofold: find efficient ways to (i) evaluate the conditional densities, (ii) sample exactly from the conditionals.

In this work, we take $\mathbb{X} = [-1, 1]^d$ and focus on sampling the multivariate Jacobi ensemble with parameters $|a^i|, |b^i| \leq 1/2$, cf. Section 2.3. We remodeled the original implementation[4] of the multivariate Jacobi ensemble sampler accompanying the work of Bardenet & Hardy (BH, 2019) in a more Python*ic* way. In particular, we address the previous challenges in the following way:

(i) contrary to BH, we leverage the Gram structure to vectorize the computations and consider (16) instead of (15), and evaluate $K_N(x, y)$ via (4) instead of (6). The overall procedure is akin to a sequential Gram-Schmidt orthogonalization of the feature vectors $\Phi(x_1), \ldots, \Phi(x_N)$. Moreover we pay special attention to avoiding unnecessary evaluations of orthogonal polynomials (OP) when computing a feature vector $\Phi(x)$. This is done by coupling the slicing feature of the Python language with the dedicated method `scipy.special.eval_jacobi`, used to evaluate the three-term recurrence relations satisfied by each of the univariate Jacobi polynomials. Given the chosen ordering $\flat$, the computation of $\Phi(x)$ requires the evaluation of $d$ recurrence relations up to depth $\sqrt[d]{N}$.

(ii) like BH, we sample each conditional in turn using a rejection sampling mechanism with the same proposal distribution. But BH take as proposal $\omega_{\mathrm{eq}}(x)\,\mathrm{d}x$, which corresponds to the limiting marginal of the multivariate Jacobi ensemble as $N$ goes to infinity; see (Simon, 2011, Section 3.11). On our side, we use a two-layer rejection sampling scheme. We rather sample from the $n$-th conditional using the marginal distribution $N^{-1}K_N(x, x)\omega(x)\,\mathrm{d}x$ as proposal and rejection constant $N/(N-(n-1))$. This allows us to reduce the number of (costly) evaluations of the acceptance ratio. The marginal distribution itself is sampled using the same proposal $\omega_{\mathrm{eq}}(x)\,\mathrm{d}x$ and rejection constant as BH. The rejection constant, of order $2^d$, is derived from Chow et al. (1994) and Gautschi (2009). We further reduced the number of OP evaluations by considering $N^{-1}K_N(x, x)\omega(x)\,\mathrm{d}x$ as a mixture, where each component in (6) involves only one OP. In the end, the expected total number of rejections is of order $2^d N \log N$. Finally, we implemented a specific rejection free method for the univariate Jacobi ensemble; a special continuous projection DPP which can be sampled exactly in $\mathcal{O}(N^2)$, by computing the eigenvalues of a random tridiagonal matrix (Killip & Nenciu, 2004, Theorem 2).

All these improvements resulted in dramatic speedups. For example, on a modern laptop, sampling a $2D$ Jacobi ensemble with $N = 1000$ points, see Figure 1(a), takes less than a minute, compared to hours previously. For more details on the sampling procedure, we refer to Appendix A.4.

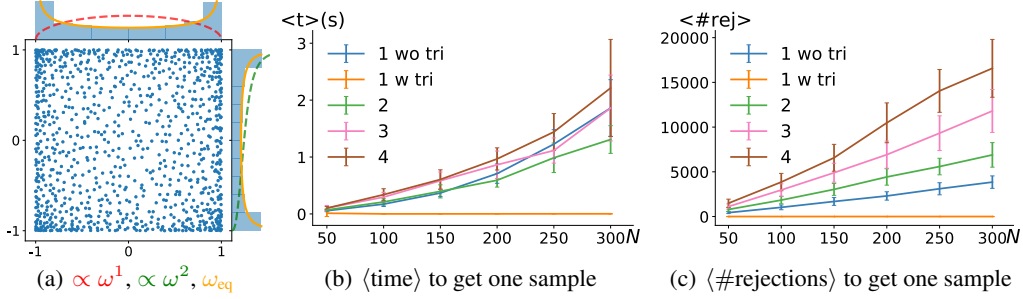

(a) $\propto \omega^1$, $\propto \omega^2$, $\omega_{\text{eq}}$      (b) $\langle \text{time} \rangle$ to get one sample      (c) $\langle \#\text{rejections} \rangle$ to get one sample

Figure 1: (a) A sample from a $2D$ Jacobi ensemble with $N = 1000$ points. (b)-(c) $a^i, b^i = -1/2$, the colors and numbers correspond to the dimension. For $d = 1$, the tridiagonal model (tri) of Killip & Nenciu offers tremendous time savings. (c) The total number of rejections grows as $2^d N \log(N)$.

## 4 Empirical investigation

We perform three main sets of experiments to investigate the properties of the BH (7) and EZ (14) estimators of the integral $\int f(x)\mu(\mathrm{d}x)$. We add the baseline vanilla Monte Carlo, where points are drawn i.i.d. proportionally to $\mu$. The two estimators are built from the multivariate Jacobi ensemble, cf. Section 2.3. First, we extend, for larger $N$, the experiments of Bardenet & Hardy (2019) illustrating their fast CLT (9) on a smooth function. Then, we illustrate Theorem 1 by considering polynomial functions that can be either fully or partially decomposed in the eigenbasis of the DPP kernel. Finally, we compare the potential of both estimators on various non smooth functions.

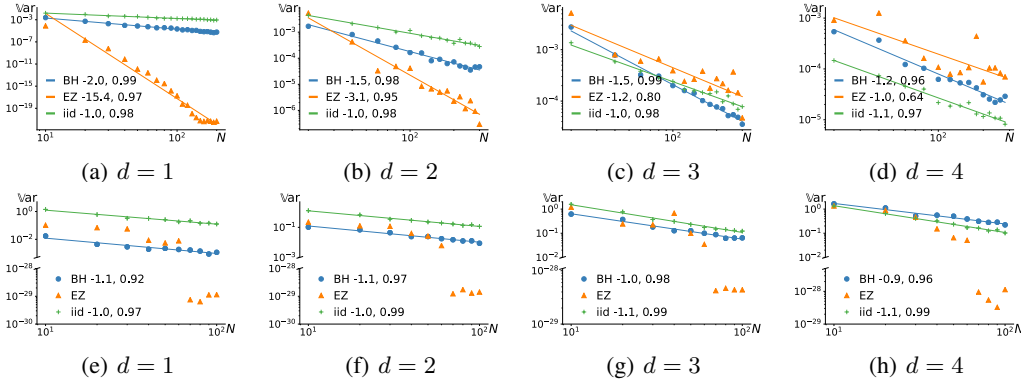

(a) $d = 1$        (b) $d = 2$        (c) $d = 3$        (d) $d = 4$

(e) $d = 1$        (f) $d = 2$        (g) $d = 3$        (h) $d = 4$

Figure 2: (a)-(d) cf. Section 4.1, the numbers in the legend are the slope and $R^2$ (e)-(h) cf. Section 4.2.

### 4.1 The bump experiment

Bardenet & Hardy (2019, Section 3) illustrate the behavior of $\widehat{I}_N^{\text{BH}}$ and its CLT (9) on a unimodal, smooth *bump* function; see Appendix B.1. The expected variance decay is of order $1/N^{1+1/d}$. We reproduce their experiment in Figure 2 for larger $N$, and compare with the behavior of $\widehat{I}_N^{\text{EZ}}$. In short, $\widehat{I}_N^{\text{EZ}}$ dramatically outperforms $\widehat{I}_N^{\text{BH}}$ in $d \leq 2$, with surprisingly fast empirical convergence rates. When $d \geq 3$, performance decreases, and $\widehat{I}_N^{\text{BH}}$ shows both faster and more regular variance decay.

To know whether we can hope for a CLT for $\widehat{I}_N^{\text{EZ}}$, we performed Kolmogorov-Smirnov tests for $N = 300$, which yielded small $p$-values across dimensions, from 0.03 to 0.24. This is compared to the same $p$-values for $\widehat{I}_N^{\text{BH}}$, which range from 0.60 to 0.99. The results are presented in Appendix B.1. The lack of normality of $\widehat{I}_N^{\text{EZ}}$ is partly due to a few outliers. Where these outliers come from is left for future work; ill-conditioning of the linear system (10) is an obvious candidate. Besides, contrary to $\widehat{I}_N^{\text{BH}}$, the estimator $\widehat{I}_N^{\text{EZ}}$ has no guarantee to preserve the sign of integrands having constant sign.

## 4.2 Integrating sums of eigenfunctions

In the next series of experiments, we are mainly interested in testing the variance decay of $\widehat{I}_N^{\mathrm{EZ}}(f)$ prescribed by Theorem 1. To that end, we consider functions of the form

$$f(x) = \sum_{\mathfrak{b}(k)=0}^{M-1} \frac{1}{\mathfrak{b}(k)+1} \phi_k(x), \tag{17}$$

whose integral w.r.t. $\mu$ is to be estimated based on realizations of the multivariate Jacobi ensemble with kernel $K_N(x,y) = \sum_{\mathfrak{b}(k)=0}^{N-1} \phi_k(x)\phi_k(y)$, where $N \neq M$ a priori. This means that the function $f$ can be either fully ($M \leq N$) or partially ($M > N$) decomposed in the eigenbasis of the kernel. In both cases, we let the number of points $N$ used to build the two estimators vary from 10 to 100 in dimensions $d = 1$ to $4$. In the first setting, we set $M = 70$. Thus, $N$ eventually reaches the number of functions used to build $f$ in (17), after what $\widehat{I}_N^{\mathrm{EZ}}$ is an exact estimator, see Figure 2(e)-(h). The second setting has $M = N + 1$, so that the number of points $N$ is never enough for the variance in (11) to be zero. The results of both settings are to be found in Appendix B.2.

In the first case, for each dimension $d$, we indeed observe a drop in the variance of $\widehat{I}_N^{\mathrm{EZ}}$ once the number of points of the DPP hits the threshold $N = M$. This is in perfect agreement with Theorem 1: once $f \in \mathcal{H}_M \subseteq \mathcal{H}_N$, the variance in (11) is zero. In the second setting, as $N$ increases the contribution of the extra mode $\phi_{\mathfrak{b}^{-1}(N)}$ in (17) decreases as $\frac{1}{N}$. Hence, from Theorem 1 we expect a variance decay of order $\frac{1}{N^2}$, which we observe in practice.

## 4.3 Further experiments

In Appendices B.3-B.6 we test the robustness of both BH and EZ estimators, when applied to functions presenting discontinuities or which do not belong to the span of the eigenfunctions of the kernel. Although the conditions of the CLT (9) associated to $\widehat{I}^{\mathrm{BH}}$ are violated, the corresponding variance decay is smooth but not as fast. For $\widehat{I}^{\mathrm{EZ}}$, the performance deteriorates with the dimension. Indeed, the cross terms arising from the Taylor expansion of the different functions introduce monomials, associated to large coefficients, that do not belong to $\mathcal{H}_N$. Sampling more points would reduce the variance (11). But more importantly, for EZ to excel, this suggests to adapt the kernel to the basis where the integrand is known to be sparse or to have fast-decaying coefficients. In regimes where BH and EZ do not shine, vanilla Monte Carlo becomes competitive for small values of $N$.

## 5 Conclusion

Ermakov & Zolotukhin (EZ, 1960) proposed a non-obvious unbiased Monte Carlo estimator using projection DPPs. It requires solving a linear system, which in turn involves evaluating both the $N$ eigenfunctions of the corresponding kernel and the integrand at the $N$ points of the DPP sample. This is yet another connection between DPPs and linear algebra. In fact, solving this linear system provides unbiased estimates of the Fourier-like coefficients of the integrand $f$ with each of the $N$ eigenfunctions of the DPP kernel. Remarkably, these estimators have identical variance, and this variance measures the accuracy of the approximation of $f$ by its projection onto these eigenfunctions. With modern arguments, we have provided a much shorter proof of these properties than in the original work of (Ermakov & Zolotukhin, 1960). Beyond this, little is known on the EZ estimator. While coming with a less interpretable variance, the more direct estimator proposed by Bardenet & Hardy (BH, 2019) has an intrinsic connection with the classical Gauss quadrature and further enjoys stronger theoretical properties when using multivariate Jacobi ensemble.

Our experiments highlight the key features of both estimators when the underlying DPP is a multivariate Jacobi ensemble, and further demonstrate the known properties of the BH estimator in yet unexplored regimes. Although EZ shows a *surprisingly fast* empirical convergence rate for $d \leq 2$, its behavior is more erratic for $d \geq 3$. Ill-conditioning of the linear system is a potential source of outliers in the distribution of the estimator. Regularization may help but would introduce a stability/bias trade-off. More generally, EZ seems worth investigating for integration or even interpolation tasks where the function is known to be decomposable in the eigenbasis of the kernel, i.e., in a setting similar to the one of Bach (2017). Finally, the new implementation of an exact sampler for multivariate Jacobi ensemble unlocks more large-scale empirical investigations and asks for more theoretical work. The associated code[1] is available in the DPPy toolbox of Gautier et al. (2019).

## Footnotes

[1] github.com/guilgautier/DPPy

[2] Many thanks to Mathieu Gerber of Univ. Bristol, UK, for digging up this result from his human memory.

[3] dppy.readthedocs.io    [4] github.com/rbardenet/dppmc

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
