[Supplementary Material]

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

# A  Methodology

## A.1  The generalized Cauchy-Binet formula: a modern argument

Johansson (2006, Section 2.2) developed a natural way to build DPPs associated to projection (potentially non-Hermitian) kernels. In this part, we focus on the generalization of the Cauchy-Binet formula (Johansson, 2006, Proposition 2.10). Its usefulness is twofold for our purpose. First, it serves to justify the fact that the normalization constant of the joint distribution (3) is one, i.e., it is indeed a probability distribution. Second, we use it as a modern and simple argument to prove a slight improvement of the result of Ermakov & Zolotukhin (1960), cf. Theorem 1. An extended version of the proof is given in Appendix A.2.

**Lemma A.** *(Johansson, 2006, Proposition 2.10) Let $(\mathbb{X}, \mathcal{B}, \mu)$ be a measurable space and consider measurable functions $\phi_0, \ldots, \phi_{N-1}$ and $\psi_0, \ldots, \psi_{N-1}$, such that $\phi_k \psi_\ell \in L^1(\mu)$. Then,*

$$\det(\langle \phi_k, \psi_\ell \rangle)_{k,\ell=0}^{N-1} = \frac{1}{N!} \int \det \mathbf{\Phi}(x_{1:N}) \det \mathbf{\Psi}(x_{1:N}) \, \mu^{\otimes N}(\mathrm{d}x), \tag{A.1}$$

*where*

$$\mathbf{\Phi}(x_{1:N}) = \begin{pmatrix} \phi_0(x_1) & \ldots & \phi_{N-1}(x_1) \\ \vdots & & \vdots \\ \phi_0(x_N) & \ldots & \phi_{N-1}(x_N) \end{pmatrix} \quad and \quad \mathbf{\Psi}(x_{1:N}) = \begin{pmatrix} \psi_0(x_1) & \ldots & \psi_{N-1}(x_1) \\ \vdots & & \vdots \\ \psi_0(x_N) & \ldots & \psi_{N-1}(x_N) \end{pmatrix}$$

## A.2  Proof of Theorem 1

First, we recall the result of Ermakov & Zolotukhin (1960), cf. Theorem 1. Then, we provide a modern proof based on the generalization of the Cauchy-Binet formula, cf. Lemma A, where we exploit the orthonormality of the eigenfunctions of the kernel.

**Theorem B.** *Consider $f \in L^2(\mu)$ and $N$ functions $\phi_0, \ldots, \phi_{N-1} \in L^2(\mu)$ orthonormal w.r.t. $\mu$, i.e.,*

$$\langle \phi_k, \phi_\ell \rangle \triangleq \int \phi_k(x)\phi_\ell(x)\mu(\mathrm{d}x) = \delta_{k\ell}, \quad \forall 0 \le k, \ell \le N-1. \tag{A.2}$$

*Let $\{\mathbf{x}_1, \ldots, \mathbf{x}_N\} \sim \mathrm{DPP}(\mu, K_N)$, with projection kernel $K_N(x,y) = \sum_{k=0}^{N-1} \phi_k(x)\phi_k(y)$. That is to say $(\mathbf{x}_1, \ldots, \mathbf{x}_N)$ has joint distribution*

$$\frac{1}{N!} \det(K_N(x_p, x_q))_{p,q=1}^N \mu^{\otimes N}(\mathrm{d}x) = \frac{1}{N!} (\det \mathbf{\Phi}(x_{1:N}))^2 \mu^{\otimes N}(\mathrm{d}x). \tag{A.3}$$

*Consider the linear system $\mathbf{\Phi}(\mathbf{x}_{1:N})y = f(\mathbf{x}_{1:N})$, that is,*

$$\begin{pmatrix} \phi_0(\mathbf{x}_1) & \ldots & \phi_{N-1}(\mathbf{x}_1) \\ \vdots & & \vdots \\ \phi_0(\mathbf{x}_N) & \ldots & \phi_{N-1}(\mathbf{x}_N) \end{pmatrix} \begin{pmatrix} y_1 \\ \vdots \\ y_N \end{pmatrix} = \begin{pmatrix} f(\mathbf{x}_1) \\ \vdots \\ f(\mathbf{x}_N) \end{pmatrix}. \tag{A.4}$$

*Then, the solution of (A.4) is unique, $\mu$-almost surely, with coordinates*

$$y_k = \frac{\det \mathbf{\Phi}_{\phi_{k-1}, f}(\mathbf{x}_{1:N})}{\det \mathbf{\Phi}(\mathbf{x}_{1:N})}, \tag{A.5}$$

*where $\mathbf{\Phi}_{\phi_{k-1}, f}(\mathbf{x}_{1:N})$ is the matrix obtained by replacing the $k$-th column of $\mathbf{\Phi}(\mathbf{x}_{1:N})$ by $f(\mathbf{x}_{1:N})$. Moreover, for all $1 \le k \le N$, the coordinate $y_k$ of the solution vector satisfies*

$$\mathbb{E}[y_k] = \langle f, \phi_{k-1} \rangle, \quad and \quad \mathbb{V}\mathrm{ar}[y_k] = \|f\|^2 - \sum_{\ell=0}^{N-1} \langle f, \phi_\ell \rangle^2. \tag{A.6}$$

*We improved the original result by showing that $\mathbb{C}\mathrm{ov}[y_j, y_k] = 0$, for all $1 \le j \ne k \le N$.*

*Proof of Theorem B.* First, the joint distribution (A.3) of $(\mathbf{x}_1, \ldots, \mathbf{x}_N)$ is proportional to $(\det \mathbf{\Phi}(x_{1:N}))^2 \mu^{\otimes N}(x)$. Thus, $\det \mathbf{\Phi}(\mathbf{x}_{1:N}) \ne 0$, $\mu$-almost surely. Hence, the matrix $\mathbf{\Phi}(\mathbf{x}_{1:N})$ defining the linear system (A.4) is invertible, $\mu$-almost surely.

The expression of the coordinates (A.5) follows from Cramer's rule.

Then, we treat the case $k = 1$, the others follow the same lines. The proof relies on Lemma A where we exploit the orthonormality of the $\phi_k$s (A.2). The expectation (A.6) reads

$$
\mathbb{E}\left[\frac{\det \boldsymbol{\Phi}_{\phi_0,f}(\mathbf{x}_{1:N})}{\det \boldsymbol{\Phi}(\mathbf{x}_{1:N})}\right] \overset{(A.3)}{=} \frac{1}{N!} \int \det \boldsymbol{\Phi}_{\phi_0,f}(x_{1:N}) \det \boldsymbol{\Phi}(x_{1:N}) \, \mu^{\otimes N}(\mathrm{d}x)
$$

$$
\overset{(A.1)}{=} \det \begin{pmatrix} \langle f, \phi_0 \rangle & (\langle f, \phi_\ell \rangle)_{\ell=1}^{N-1} \\ (\langle \phi_k, \phi_0 \rangle)_{k=1}^{N-1} & (\langle \phi_k, \phi_\ell \rangle)_{k,\ell=1}^{N-1} \end{pmatrix}
$$

$$
\overset{(A.2)}{=} \det \begin{pmatrix} \langle f, \phi_0 \rangle & (\langle f, \phi_\ell \rangle)_{\ell=1}^{N-1} \\ 0_{N-1,1} & I_{N-1} \end{pmatrix}
$$

$$
= \langle f, \phi_0 \rangle. \tag{A.7}
$$

Similarly, the second moment reads

$$
\mathbb{E}\left[\left(\frac{\det \boldsymbol{\Phi}_{\phi_0,f}(\mathbf{x}_{1:N})}{\det \boldsymbol{\Phi}(\mathbf{x}_{1:N})}\right)\right] \overset{(A.3)}{=} \frac{1}{N!} \int \det \boldsymbol{\Phi}_{\phi_0,f}(x_{1:N}) \det \boldsymbol{\Phi}_{\phi_0,f}(x_{1:N}) \, \mu^{\otimes N}(\mathrm{d}x)
$$

$$
\overset{(A.1)}{=} \det \begin{pmatrix} \langle f, f \rangle & (\langle f, \phi_\ell \rangle)_{\ell=1}^{N-1} \\ (\langle \phi_k, f \rangle)_{k=1}^{N-1} & (\langle \phi_k, \phi_\ell \rangle)_{k,\ell=1}^{N-1} \end{pmatrix}
$$

$$
\overset{(A.2)}{=} \det \begin{pmatrix} \|f\|^2 & (\langle f, \phi_\ell \rangle)_{\ell=1}^{N-1} \\ (\langle f, \phi_k \rangle)_{k=1}^{N-1} & I_{N-1} \end{pmatrix}
$$

$$
= \|f\|^2 - \sum_{k=1}^{N-1} \langle f, \phi_k \rangle^2. \tag{A.8}
$$

Finally, the variance in (A.6) = (A.8) - (A.7)$^2$.

With the same arguments, for $j \neq k$, we can compute the covariance $\mathbb{Cov}[y_j, y_k]$. For simplicity, we treat only the case $j = 1, k = 2$, the general case follows the same lines.

$$
\mathbb{Cov}[y_1, y_2] = \mathbb{E}\left[\frac{\det \boldsymbol{\Phi}_{\phi_0,f}(\mathbf{x}_{1:N})}{\det \boldsymbol{\Phi}(\mathbf{x}_{1:N})} \frac{\det \boldsymbol{\Phi}_{\phi_1,f}(\mathbf{x}_{1:N})}{\det \boldsymbol{\Phi}(\mathbf{x}_{1:N})}\right] - \mathbb{E}\left[\frac{\det \boldsymbol{\Phi}_{\phi_0,f}(\mathbf{x}_{1:N})}{\det \boldsymbol{\Phi}(\mathbf{x}_{1:N})}\right] \mathbb{E}\left[\frac{\det \boldsymbol{\Phi}_{\phi_1,f}(\mathbf{x}_{1:N})}{\det \boldsymbol{\Phi}(\mathbf{x}_{1:N})}\right]
$$

$$
\overset{(A.3),(A.7)}{=} \frac{1}{N!} \int \det \boldsymbol{\Phi}_{\phi_0,f}(x_{1:N}) \det \boldsymbol{\Phi}_{\phi_1,f}(x_{1:N}) \, \mu^{\otimes N}(\mathrm{d}x) - \langle f, \phi_0 \rangle \langle f, \phi_1 \rangle
$$

$$
\overset{(A.1)}{=} \det \begin{pmatrix} \langle f, \phi_0 \rangle & \langle f, f \rangle & (\langle f, \phi_\ell \rangle)_{\ell=2}^{N-1} \\ (\langle \phi_k, \phi_0 \rangle)_{k=1}^{N-1} & (\langle \phi_k, f \rangle)_{k=1}^{N-1} & (\langle \phi_k, \phi_\ell \rangle)_{k=1,\ell=2}^{N-1} \end{pmatrix} - \langle f, \phi_0 \rangle \langle f, \phi_1 \rangle
$$

$$
\overset{(A.2)}{=} \det \begin{pmatrix} \langle f, \phi_0 \rangle & \|f\|^2 & (\langle f, \phi_\ell \rangle)_{\ell=2}^{N-1} \\ 0 & \langle \phi_1, f \rangle & 0 \\ 0_{N-2,1} & (\langle \phi_k, f \rangle)_{k=2}^{N-1} & I_{N-2} \end{pmatrix} - \langle f, \phi_0 \rangle \langle f, \phi_1 \rangle
$$

$$
= \langle f, \phi_0 \rangle \langle f, \phi_1 \rangle - \langle f, \phi_0 \rangle \langle f, \phi_1 \rangle = 0.
$$

$\square$

## A.3 The EZ estimator as a quadrature rule

In this part, we consider Theorem B in the setting where one of the eigenfunctions of the kernel, say $\phi_0$ is constant. In this case, we show that $\widehat{I}_N^{\mathrm{EZ}}(f)$ defined by (14) provides an unbiased estimate of $\int_{\mathbb{X}} f(x)\mu(\mathrm{d}x)$ with known variance. In addition, it can be seen as a quadrature rule in the sense of (1), with weights a priori non negative weights $\omega_n$ that sum to $\mu(\mathbb{X})$. This is a non obvious fact, judging from the expression (14) of the estimator.

**Proposition 1.** *Consider $\phi_0$ constant in Theorem [B]. Then, solving the corresponding linear system* ([A.4]) *allows to construct*

$$\widehat{I}_N^{EZ}(f) \triangleq \frac{y_1}{\phi_0} = \mu(\mathbb{X})^{1/2} \frac{\det \mathbf{\Phi}_{\phi_0,f}(\mathbf{x}_{1:N})}{\det \mathbf{\Phi}(\mathbf{x}_{1:N})}, \tag{A.9}$$

*as an unbiased estimator of $\int_{\mathbb{X}} f(x)\mu(\mathrm{d}x)$, with variance equal to $\mu(\mathbb{X})\times$([A.6]). In addition, it can be seen as a random quadrature rule ([1]) with weights summing to $\mu(\mathbb{X})$.*

*Proof.* Since $\phi_0$ is constant with unit norm we have $\phi_0 = \mu(\mathbb{X})^{-1/2}$, so that

$$\mathbb{E}\Big[\widehat{I}_N^{EZ}(f)\Big] = \frac{1}{\phi_0}\mathbb{E}[y_1] = \frac{1}{\phi_0}\langle f, \phi_0 \rangle = \int_{\mathbb{X}} f(x)\,\mathrm{d}x,$$

and

$$\mathbb{V}\mathrm{ar}\Big[\widehat{I}_N^{EZ}(f)\Big] = \frac{1}{\phi_0^2}\mathbb{V}\mathrm{ar}[y_1] = \mu(\mathbb{X}) \times \text{(A.6)}.$$

In addition, ([A.9]) can be written

$$\widehat{I}_N^{EZ}(f) = \mu(\mathbb{X})^{1/2} \frac{\det \mathbf{\Phi}_{\phi_0,f}(\mathbf{x}_{1:N})}{\phi_0 \det \mathbf{\Phi}_{\phi_0,1}(\mathbf{x}_{1:N})} = \mu(\mathbb{X}) \frac{\det \mathbf{\Phi}_{\phi_0,f}(\mathbf{x}_{1:N})}{\det \mathbf{\Phi}_{\phi_0,1}(\mathbf{x}_{1:N})},$$

and the expansion of the numerator w.r.t. the first column yields

$$\widehat{I}_N^{EZ}(f) = \sum_{n=1}^{N} f(\mathbf{x}_n) \underbrace{\frac{\mu(\mathbb{X})}{\det \mathbf{\Phi}_{\phi_0,1}(\mathbf{x}_{1:N})}(-1)^{1+n}\det(\phi_k(x_p))_{k=1,p=1\neq n}^{N-1,N}}_{\triangleq \omega_n(\mathbf{x}_{1:N})}. \tag{A.10}$$

Note that there is a priori no reason for the weights to be nonnegative. Finally,

$$\sum_{n=1}^{N} \omega_n(\mathbf{x}_{1:N}) = \frac{\mu(\mathbb{X})}{\cancel{\det \mathbf{\Phi}_{\phi_0,1}(\mathbf{x}_{1:N})}} \underbrace{\sum_{n=1}^{N}(-1)^{1+n}\det(\phi_k(x_p))_{k=1,p=1\neq n}^{N-1,N}}_{=\cancel{\det \mathbf{\Phi}_{\phi_0,1}(\mathbf{x}_{1:N})}} = \mu(\mathbb{X}).$$

This concludes the proof. $\qquad\square$

## A.4 Sampling from the multivariate Jacobi ensemble

We mention that the code[1] and the documentation[3] associated to this work are available in the DPPy toolbox of Gautier et al. (2019).

In dimension $d = 1$, we implemented the random tridiagonal matrix model of Killip & Nenciu (2004, Theorem 2) to sample from the univariate Jacobi ensemble, with base measure $\mu(\mathrm{d}x) = (1-x)^a(1+x)^b\,\mathrm{d}x$, where $a, b > -1$. That is to say, this one dimensional continuous projection DPP with $N$ points can be sampled in $\mathcal{O}(N^2)$, by computing the eigenvalues of random tridiagonal matrix with i.i.d. coefficients of size $N \times N$.

Next, for $d \geq 2$, we detail the procedure described in Section [3.3] for sampling exactly from the multivariate Jacobi ensemble with parameters $|a^i|, |b^i| \leq \frac{1}{2}$, for all $1 \leq i \leq d$.

More specifically, we consider sampling exactly from the projection $\mathrm{DPP}(\mu, K_N)$ where

- $\mu(\mathrm{d}x) = \omega(x)\,\mathrm{d}x$, with

$$\omega(x) = \prod_{i=1}^{d}\omega^i(x^i), \quad \text{where} \quad \omega^i(z) = \prod_{i=1}^{d}(1-z)^{a^i}(1+z)^{b^i}, \quad \text{and} \quad |a^i|, |b^i| \leq \frac{1}{2}. \tag{A.11}$$

- $K_N(x,y) = \sum_{\mathfrak{b}(b)=0}^{N-1}\phi_k(x)\phi_k(y)$, with

$$\phi_k(x) = \prod_{i=1}^{d}\phi_{k^i}^i(x^i), \quad \text{where} \quad \int_{-1}^{1}\phi_u^i(z)\phi_v^i(z)\omega^i(z)\,\mathrm{d}z = \delta_{uv}. \tag{A.12}$$

Figure A.1: (middle) a sample of a 2D Jacobi ensemble with $N = 1000$ points. The normalized reference densities, proportional to $(1-x)^{a^1}(1+x)^{b^1}$ and $(1-y)^{a^2}(1+y)^{b^2}$, are displayed in dashed lines. The empirical marginal densities which converges to the arcsine density $\omega_{\text{eq}}(x) = \frac{1}{\pi\sqrt{1-x^2}}$ is plotted in solid line. Then we plot the same sample where the disk centered at $\mathbf{x}_n$ has now an area proportional to: (left) the weight $1/K_N(\mathbf{x}_n, \mathbf{x}_n)$ of $\widehat{I}_N^{\text{BH}}(f)$ in (7), observe that these weights serve as a proxy for the reference measure, like Gaussian quadrature (right) the weight of $\widehat{I}_N^{\text{EZ}}(f)$ given by (A.10), observe that they can be either positive or negative. The histogram of the absolute value of the weights is plotted on the marginal axes

As an illustration, Figure A.1 displays a sample of a $d = 2$ Jacobi ensemble with $N = 1000$ points. Our sampling scheme is an instance of the generic chain-rule-based procedure of Hough et al. (2006, Algorithm 18) where the knowledge of the eigenfunctions can be leveraged, see also Lavancier et al. (2012, Algorithm 1). In our case, sampling $N$ points in dimension $d$, requires an expected total number of rejections of order $2^d N \log(N)$. As mentioned in Section 3.3, to sample from $\{\mathbf{x}_1, \ldots, \mathbf{x}_N\} \sim \text{DPP}(\mu, K_N)$ it is enough to sample $(\mathbf{x}_1, \ldots, \mathbf{x}_N)$ and forget the order the points were selected. Starting from the two formulations (3) and (5) of the joint distribution, the chain rule scheme can be derived from two different perspectives. Either by expressing the determinant $\det(K_N(x_p, x_n))_{p,n=1}^N$ using Schur complements

$$(3) = \frac{1}{N!} \det(K_N(x_p, x_n))_{p,n=1}^N \prod_{n=1}^N \omega(x_n)\,\mathrm{d}x_n \tag{A.13}$$

$$= \frac{K_N(x_1, x_1)}{N}\omega(x_1)\,\mathrm{d}x_1 \prod_{n=2}^N \omega(x_n)\,\mathrm{d}x_n \frac{K_N(x_n, x_n) - \mathbf{K}_{n-1}(x_n)^\mathsf{T}\mathbf{K}_{n-1}^{-1}\mathbf{K}_{n-1}(x_n)}{N - (n-1)}\omega(x_n)\,\mathrm{d}x_n,$$

where $\mathbf{K}_{n-1}(\cdot) = (K_N(x_1, \cdot), \ldots, K_N(x_{n-1}, \cdot))^\mathsf{T}$, and $\mathbf{K}_{n-1} = (K_N(x_p, x_q))_{p,q=1}^{n-1}$. Or geometrically using the base×height formula to express $(\det \mathbf{\Phi}(x_{1:N}))^2$ as the squared volume of the parallelotope spanned by $\mathbf{\Phi}(x_1), \ldots, \mathbf{\Phi}(x_N)$

$$(5) = \frac{1}{N!} \text{volume}^2(\mathbf{\Phi}(x_1), \ldots, \mathbf{\Phi}(x_N)) \prod_{n=1}^N \omega(x_n)\,\mathrm{d}x_n$$

$$= \frac{\|\mathbf{\Phi}(x_1)\|^2}{N}\omega(x_1)\,\mathrm{d}x_1 \prod_{n=2}^N \frac{\text{distance}^2(\mathbf{\Phi}(x_n), \text{span}\{\mathbf{\Phi}(x_p)\}_{p=1}^{n-1})}{N - (n-1)}\omega(x_n)\,\mathrm{d}x_n. \tag{A.14}$$

Note that, contrary to (A.14), the formulation (A.13) does not require a priori knowledge of the eigenfunctions of the projection kernel $K_N$.

Like Bardenet & Hardy (2019), we sample each conditional in turn using rejection sampling with the same proposal distribution and rejection bound. But where Bardenet & Hardy (2019) use the formulation (A.13) of the chain rule we consider the geometrical perspective (A.14). This allows for a implementation that is simpler (no need to update $\mathbf{K}_{n-1}^{-1}$), fully vectorized, and more interpretable: akin to a sequential Gram-Schmidt orthogonalization of the feature vectors $\mathbf{\Phi}(x_1), \ldots, \mathbf{\Phi}(x_N)$.

Moreover, contrary to Bardenet & Hardy (2019) who take $\omega_{\text{eq}}(x)\,\mathrm{d}x$ as proposal to sample from the each of the conditionals, we use a two-layer rejection sampling scheme. We rather sample from the $n$-th conditional using the marginal distribution $N^{-1}K_N(x, x)\omega(x)\,\mathrm{d}x$. This choice of proposal allows us to reduce the number of (costly) evaluations of the acceptance ratio.

The rejection constant associated to the $n$-th conditional in (A.13) reads

$$\frac{(N-(n-1))^{-1}\big(K_N(x,x)-\mathbf{K}_{n-1}(x)^\intercal\mathbf{K}_{n-1}^{-1}\mathbf{K}_{n-1}(x)\big)\omega(x)}{N^{-1}K_N(x,x)\omega(x)}$$

$$=\frac{N}{N-(n-1)}\frac{K_N(x,x)-\mathbf{K}_{n-1}(x)^\intercal\mathbf{K}_{n-1}^{-1}\mathbf{K}_{n-1}(x)}{K_N(x,x)}\leq\frac{N}{N-(n-1)}. \tag{A.15}$$

The marginal distribution itself is sampled using the same proposal $\omega_{\text{eq}}(x)\,\mathrm{d}x$ and rejection constant as Bardenet & Hardy (2019). However, we further reduce the number of computations by considering $N^{-1}K_N(x,x)\omega(x)\,\mathrm{d}x$ as a mixture, see Section A.4.1

### A.4.1 Generate samples from the marginal distribution

First, observe that the marginal density can be written as a mixture of $N$ probability densities where each component is assigned the same weight $1/N$

$$\frac{1}{N}K_N(x,x)\omega(x)=\frac{1}{N}\sum_{\mathfrak{b}(k)=0}^{N-1}\phi_k(x)^2\omega(x). \tag{A.16}$$

Thus, sampling from (A.16) can be done in two steps:

(i) select a multi-index $k=\mathfrak{b}^{-1}(n)$ with $n$ drawn uniformly at random in $\{0,\dots,N-1\}$

(ii) sample from $\phi_k(x)^2\omega(x)\,\mathrm{d}x$

We perform Step (ii) using rejection sampling with proposal distribution

$$\omega_{\text{eq}}(x)\,\mathrm{d}x=\prod_{i=1}^d\frac{1}{\pi\sqrt{1-(x^i)^2}}\,\mathrm{d}x^i, \tag{A.17}$$

which corresponds to the limiting marginal distribution of the multivariate Jacobi ensemble as $N$ goes to infinity; see (Simon, 2011, Section 3.11) and Figure A.1. The acceptance ratio writes

$$\frac{\phi_k(x)^2\omega(x)}{\omega_{\text{eq}}(x)}\overset{\text{(A.12)(A.11)}}{\underset{\text{(A.17)}}{=}}\prod_{i=1}^d\frac{\phi_{k^i}^i(x^i)^2\times(1-x^i)^{a^i}(1+x^i)^{b^i}}{\pi^{-1}(1-x^i)^{-\frac{1}{2}}(1+x^i)^{-\frac{1}{2}}}$$

$$=\prod_{i=1}^d\pi(1-x^i)^{a^i+\frac{1}{2}}(1+x^i)^{b^i+\frac{1}{2}}\phi_{k^i}^i(x^i)^2. \tag{A.18}$$

Each of the terms that appear in (A.18) can be bounded using the following recipe:

(a) For $k^i=0$, $\phi_0^i$ is constant and the orthonormality w.r.t. $(1-x)^{a^i}(1+x)^{b^i}\,\mathrm{d}x$ yields

$$(\phi_0^i)^2\int_{-1}^1(1-x)^{a^i}(1+x)^{b^i}\,\mathrm{d}x=1\iff(\phi_0^i)^2=\frac{1}{2^{a^i+b^i+1}B(a^i+1,b^i+1)}, \tag{A.19}$$

so that the corresponding term in (A.18) becomes

$$\frac{\pi(1-x)^{a^i+\frac{1}{2}}(1+x)^{b^i+\frac{1}{2}}}{2^{a^i+b^i+1}B(a^i+1,b^i+1)}\leq\frac{\pi(1-m)^{a^i+\frac{1}{2}}(1+m)^{b^i+\frac{1}{2}}}{2^{a^i+b^i+1}B(a^i+1,b^i+1)}\triangleq C_{k^i=0}\leq2, \tag{A.20}$$

where $m=\underset{-1\leq x\leq1}{\operatorname{argmax}}(1-x)^{a^i+\frac{1}{2}}(1+x)^{b^i+\frac{1}{2}}=\begin{cases}0,&\text{if }a^i=b^i=-\frac{1}{2},\\\frac{b^i-a^i}{a^i+b^i+1},&\text{otherwise.}\end{cases}$

(b) For $k^i\geq1$, we use the bound $C_{k^i\geq1}$ (A.22) provided originally by Chow et al. (1994). As mentioned by Gautschi (2009), this bound is probably maximal for $k^i=1$ and parameters $a^i\approx-0.0691,b^i=1/2$, with value $\approx0.64297807\pi\approx2.02$.

Finally, the expected number of rejections to perform Step (ii) is equal to $\prod_{i=1}^{d} C_{k^i}$ which is of order $2^d$, and the expected total number of rejections of the chain rule (A.13) is of order

$$\sum_{n=1}^{N} 2^d \frac{N}{N - (n-1)} = 2^d N \sum_{n=1}^{N} \frac{1}{n} \approx 2^d N \log(N). \tag{A.21}$$

**Proposition 2.** *(Gautschi, 2009, Equation 1.3) Let $(\phi_k)_{k \geq 0}$ be the (univariate) orthonormal polynomials w.r.t. $(1-x)^a (1+x)^b \, \mathrm{d}x$ with $|a| \leq \frac{1}{2}, |b| \leq \frac{1}{2}$. Then, for any $x \in [-1, 1]$ and $k \geq 1$,*

$$\pi (1-x)^{a+\frac{1}{2}} (1+x)^{b+\frac{1}{2}} \phi_k(x)^2 \leq \frac{2 \, \Gamma(k+a+b+1) \, \Gamma(k + \max(a,b) + 1)}{k! \, (k + \frac{a+b+1}{2})^{2\max(a,b)} \, \Gamma(k + \min(a,b) + 1)}. \tag{A.22}$$

### A.4.2  Empirical timing and number of rejections

In Figure A.2 we illustrate the following observations. Computing the acceptance ratio (A.15) requires to propagate the recurrence relations up to order $\sqrt[d]{N}$. Thus, for a given number of points $N$, the larger the dimension, the smaller the depth of the recurrence. This could hint that, evaluating the kernel (6) becomes cheaper as $d$ increases. However, the rejection rate also increases, so that in practice, it is not cheaper to sample in larger dimensions because the number of rejections dominates. In the particular case of dimension $d = 1$, samples are generated using the fast and rejection-free tridiagonal matrix model of Killip & Nenciu (2004, Theorem 2). This grants huge time savings compared to the acceptance-rejection method.

Finally, some remarks are in order. Sampling from the $n$-th conditional distribution using rejection sampling is common practice (Lavancier et al., 2012, Section 2.4.2). However, tailored proposals with tight rejection constants are required (Lavancier et al., 2012, Appendices E-F). Taking the marginal distribution $N^{-1} K_N(x, x) \omega(x) \, \mathrm{d}x$ as proposal yields a $N/(N - (n-1))$ rejection constant and applies in the general case. Nevertheless, it remains to sample from this marginal distribution Rejection sampling might be a first option to sample from $N^{-1} K_N(x, x) \omega(x) \, \mathrm{d}x$, but when the eigenfunctions are available it could be another option to see it as a mixture (cf. Section A.4.1), where good proposals for each $\phi_k(x)^2 w(x) \, \mathrm{d}x$ are required.

In the case of (multivariate) orthogonal polynomial ensembles (cf. Section 2.3), evaluations of $K_N(x, y)$ (6) can be performed using the Gram representation (4), $K_N(x, y) = \Phi(x)^\top \Phi(y)$ and one can leverage the three-term recurrence relations satisfied by each of the univariate Jacobi polynomials $(\phi_\ell^i)_\ell$. This is what we do in our special case, we use the dedicated function `scipy.special.eval_jacobi` to evaluate, up to depth $\sqrt[d]{N}$, the three-term recurrence relations satisfied by each of the univariate Jacobi. Instead of calling the recursive routine internally to evaluate $\Phi(x)$, the corresponding $d \sqrt[d]{N}$ univariate polynomials or $N$ multivariate polynomials could be stored in some way and evaluated pointwise on the fly. The preprocessing time and the memory required would increase but it might accelerate the evaluation of $\Phi(x)$.

(a) ⟨time⟩ to get one sample

(b) ⟨#rejections⟩ to get one sample

Figure A.2: $a^i, b^i = -1/2$, the colors and numbers correspond to the dimension. For $d = 1$, the tridiagonal model (tri) of Killip & Nenciu offers tremendous time savings. (b) The total number of rejections grows as $2^d N \log(N)$ (A.21).

# B Experiments

## B.1 Reproducing the bump example

In Section 4.1, we reproduce the experiment of Bardenet & Hardy (2019, Section 3) where they illustrate the behavior of $\widehat{I}_N^{\mathrm{BH}}$ on a unimodal, smooth bump function:

$$f(x) = \prod_{i=1}^{d} \exp\left(-\frac{1}{1 - \varepsilon - (x^i)^2}\right) \mathbb{1}_{(-\sqrt{1-\varepsilon}, \sqrt{1-\varepsilon})}(x^i). \tag{B.1}$$

We take $\varepsilon = 0.05$. For each value of $N$, we sample 100 times from the same multivariate Jacobi ensemble with i.i.d. uniform parameters on $[-1/2, 1/2]$, compute the resulting 100 values of each estimator, and plot the two resulting sample variances. In addition, in Figure B.2 we test the potential hope for a CLT for $\widehat{I}_N^{\mathrm{EZ}}$ and compare with $\widehat{I}_N^{\mathrm{BH}}$ for which the CLT (9) holds, in the regime $N = 300$.

(a) $d = 1$

(b) $d = 2$

(c) $d = 3$

(d) $d = 4$

Figure B.1: Reproducing the bump function ($\varepsilon = 0.05$) experiment of Bardenet & Hardy (2019), cf. Section 4.1. Observe the expected variance decay of order $1/N^{1+1/d}$ for BH. Although vanilla Monte Carlo becomes competitive for small $N$ as $d$ increases, its variance decay is of order $1/N \geq 1/N^{1+1/d}$. Thus, there will always be meeting point, for some $N^*$, after which the variance of BH will be smaller. For $d = 1$, EZ has almost no variance for $N \geq 100$: the bump function is extremely well approximated by a polynomials of degree $N \geq 100$.

(a) $d = 1$

(b) $d = 2$

(c) $d = 3$

(d) $d = 4$

Figure B.2: Histogram of 100 independent estimates $\widehat{I}_N^{\text{BH}}$ and $\widehat{I}_N^{\text{EZ}}$ of the integral of the bump function ($\varepsilon = 0.05$) with $N = 300$ and associated p-value of Kolmogorov-Smirnov test, cf. Section 4.1. The fluctuations of BH confirm to be Gaussian (cf. CLT (9)). (a) the bump function is extremely well approximated by a polynomial of degree 300 hence $\widehat{I}_N^{\text{EZ}}$ has almost no variance. (b)-(c)-(d) A few outliers seem to break the potential Gaussianity of $\widehat{I}_N^{\text{EZ}}(f)$. (d) $\widehat{I}_N^{\text{EZ}}(f)$ does not preserve the sign of the integrand.

## B.2 Integrating sums of eigenfunctions

Figure B.3 gives the results of the first setting set in Section 4.2, where we integrate a sum of $M = 70$ kernel eigenfunctions. In this case, $EZ$ has zero variance once $N \geq M$, a performance that can be reached neither by BH nor vanilla Monte Carlo.

Figure B.4 illustrates the second setting, where the sum always has one more eigenfunction than there are points in the DPP samples. In this case, the conditions for the CLT of BH, cf (9), are not met; there is no $1/N^{1+1/d}$ guarantee on the variance decay for BH estimator. The performance of BH and vanilla Monte Carlo are comparable. By construction, the variance of EZ decays as $1/N^2 \leq 1/N$. Thus, there will always be meeting point, for some $N^*$, after which the variance of EZ will be smaller than vanilla Monte Carlo.

(a) $d = 1$

(b) $d = 2$

(c) $d = 3$

(d) $d = 4$

Figure B.3: Comparison of $\widehat{I}_N^{\mathrm{BH}}$ and $\widehat{I}_N^{\mathrm{EZ}}$ integrating a finite sum of $70$ eigenfunctions of the DPP kernel as in (17), cf. Section 4.2.

(a) $d = 1$

(b) $d = 2$

(c) $d = 3$

(d) $d = 4$

Figure B.4: Comparison of $\widehat{I}_N^{\mathrm{BH}}$ and $\widehat{I}_N^{\mathrm{EZ}}$ for a linear combination of $N + 1$ eigenfunctions of the DPP kernel as in (17), cf. Section 4.2.

We now consider cases where the guarantees of BH not EZ are unknown.

## B.3  Integrating absolute value

We consider estimating the integral

$$\int_{[-1,1]^d} \prod_{i=1}^d |x^i|(1-x^i)^{a^i}(1+x^i)^{b^i} \, \mathrm{d}x^i \tag{B.2}$$

where $a^1, b^1 = -\frac{1}{2}$ and $a^i, b^i$ i.i.d. uniformly in $[-\frac{1}{2}, \frac{1}{2}]$, using BH (7) and EZ (14) estimators.

Results are given in Figure B.5. In dimension $d = 1$, the absolute value is well approximated by its truncated Taylor series of low order and EZ performs very well, but as the dimension increases, its performance is more erratic. For $d \leq 2$, the performance of BH is smooth and better that vanilla Monte Carlo. In particular, for $d \leq 2$, the rate $1/N^{1+1/d}$ seems to hold for BH while the conditions for the CLT (9) are not satisfied. But it seems no longer true in larger dimension.

(a) $d = 1$

(b) $d = 2$

(c) $d = 3$

(d) $d = 4$

Figure B.5: Comparison of $\widehat{I}_N^{\mathrm{BH}}$ and $\widehat{I}_N^{\mathrm{EZ}}$ for absolute value, cf. Section 4.3.

## B.4 Integrating Heaviside

Let $H(x) = \begin{cases} 1, & \text{if } x > 0 \\ 0, & \text{otherwise} \end{cases}$. We consider estimating the integral

$$\int_{[-1,1]^d} \prod_{i=1}^{d} 2\left(H(x^i) - \frac{1}{2}\right)(1 - x^i)^{a^i}(1 + x^i)^{b^i} \, \mathrm{d}x^i \tag{B.3}$$

where $a^1, b^1 = -\frac{1}{2}$ and $a^i, b^i$ i.i.d. uniformly in $[-\frac{1}{2}, \frac{1}{2}]$, using BH (7) and EZ (14) estimators.

Results are given in Figure B.6. The EZ estimator behaves in a very erratic way; it does not seem robust to the discontinuity we have introduced. This can be explained by considering $H(x) = \frac{1}{2}\lim_{\epsilon \to 0} 1 + \tanh \frac{x}{\epsilon}$ and taking the product of the Taylor series expansions of $\tanh$; the square of the coefficients in front of the monomials in such expansion become very large as $\epsilon \to 0$. One could expect better behavior for very large $N$. The performance of BH is smooth and the rate $1/N^{1+1/d}$ seems to hold despite the conditions for the CLT (9) are not satisfied.

(a) $d = 1$

(b) $d = 2$

(c) $d = 3$

(d) $d = 4$

Figure B.6: Comparison of $\widehat{I}_N^{\mathrm{BH}}$ and $\widehat{I}_N^{\mathrm{EZ}}$ for Heaviside function, cf. Section 4.3.

## B.5 Integrating cosine

We consider estimating the integral

$$\int_{[-1,1]^d} \prod_{i=1}^d \cos(\pi x^i)(1-x^i)^{a^i}(1+x^i)^{b^i} \, \mathrm{d}x^i \tag{B.4}$$

where $a^1, b^1 = -\frac{1}{2}$ and $a^i, b^i$ i.i.d. uniformly in $[-\frac{1}{2}, \frac{1}{2}]$, using BH (7) and EZ (14) estimators.

Results are given in Figure B.7 The EZ estimator behaves well for $d \leq 2$ but its performance deteriorates for $d \geq 3$. Indeed, the cross terms arising from the Taylor expansion of the different $\cos(\pi x^i)$ introduce monomials, associated to large coefficients, that do not belong to $\mathcal{H}_N$. One could expect better behavior for very large $N$. For $d \leq 2$, the rate $1/N^{1+1/d}$ for BH seems to hold despite the conditions for the CLT (9) are not satisfied. For $d \geq 3$, BH and vanilla Monte Carlo behave similarly.

(a) $d = 1$

(b) $d = 2$

(c) $d = 3$

(d) $d = 4$

Figure B.7: Comparison of $\widehat{I}_N^{\mathrm{BH}}$ and $\widehat{I}_N^{\mathrm{EZ}}$ for cosine, cf. Section 4.3.

## B.6 Integrating a mixture of smooth and non smooth functions

Let $f(x) = H(x)(\cos(\pi x) + \cos(2\pi x) + \sin(5\pi x))$. We consider estimating the integral

$$\int_{[-1,1]^d} \prod_{i=1}^{d} f(x^i)(1-x^i)^{a^i}(1+x^i)^{b^i} \, \mathrm{d}x^i \tag{B.5}$$

where $a^1, b^1 = -\frac{1}{2}$ and $a^i, b^i$ i.i.d. uniformly in $[-\frac{1}{2}, \frac{1}{2}]$, using BH (7) and EZ (14) estimators.

(a) $d = 1$

(b) $d = 2$

(c) $d = 3$

(d) $d = 4$

Figure B.8: Comparison of $\widehat{I}_N^{\mathrm{BH}}$ and $\widehat{I}_N^{\mathrm{EZ}}$, cf. Section 4.3.