[Reviews · NeurIPS 2019]

Reviewer 1



Update: Thanks for the detailed feedback. I appreciate the theoretical and methodological exploration and improvements, but I hope the presentation can be made more clear to eliminate possible confusions. ====== I believe the quality of the paper is good, as the technical development and proof are sound and theoretical concerns are widely covered. I am not satisfied with the presentation. I cannot distinguish clearly the novel contributions from existing results, e.g., I cannot tell if Theorem 1 has new results or is a known result listed for a modern proof, if it is a novel idea to use the multivariate orthonormal polynomials in EZ, and if Section 3.1 contains novel results. I also expect more details on the methodological modification over BH for the proposed sampling method for multivariate Jacobi ensemble. I expect at least one experimental result is provided in the main context for each of sections 4.2 and 4.3. Based on the confusion, I cannot tell precisely the originality and significance. But I think a modern analysis on an old method in parallel with a recent one aligns important methods in the field, providing more connections and comparisons of the two methods, and making a foundation for more methods and analysis in the field.

Reviewer 2



I believe that the paper contains extremely interesting material. It is very interesting from a theoretical and practical point of view. Quadrature techniques (deterministic or stochastic) are fundamental tools for several applications. However, in my opinion, the key- underlying point of the ideas contained in the paper is the procedure (an efficient one) for drawing from DPPs. If there is one sampling scheme described in the current version of the paper, it must be explained better and explicitly remarked with a table, for instance. If the authors consider that is just a background part contained in other previous papers, I disagree. I believe this is the core, the key point, in order to use DPPs then it should appear in a background section at least. Moreover, this part should be very clean and clear in order to guide and motivate the reader inside the more complex and theoretical parts of your contribution. If the reader acquires the ability of generating a realization of a DPP, surely your paper increases its impact. This is also true that the provided code helps substantially in this sense.

Reviewer 3



Determinantal point processes (DPP) are currently a very active research topic. They provide a law on point processes that favors repulsion. The use of DPP in Monte Carlo methods is tempting, but rather difficult because simulation algorithms are not obvious. The present paper studies two Monte Carlo algorithms based on DPP: Ermakov & Zolotukhin (EZ, 1960) and Bardenet & Hardy (2016, BH). The latter paper provides a fast CLT of the estimator, while there is no such results about the EZ estimate in the literature. After proving the consistency of the EZ estimate via DPP, the paper discusses the possibility of a normal asymptotic behavior on simulation. DPP based Monte Carlo algorithms are certainly new and might represent a major improvement in this domain, since repulsion induces a faster speed of convergence than iid sampling or importance sampling or Markov chain Monte Carlo. The present paper highlights the key features of the EZ and BH estimators based on DPP.

[Author Response · NeurIPS 2019]

<sub>1</sub> We thank reviewers **R1**, **R2** and **R3** for their constructive comments. We give here answers to the main ones.

<sub>2</sub> **R2** *If there is one sampling scheme [...] it must be explained better [...] this part should be very clean and clear.*

<sub>3</sub> **R1** *I also expect more details on the methodological modification over BH for the proposed sampling method.*

<sub>4</sub> We agree, and we have expanded Section 3.3 and Appendix A.4 to shed light on $(i)$ sampling continuous DPPs, and $(ii)$
<sub>5</sub> our contributions that led to cutting sampling time by orders of magnitude for our specific DPPs. The following is a
<sub>6</sub> short preview of the improved presentation of our contributions on the sampling procedure.

<sub>7</sub> The original sampling algorithm for projection DPPs by Hough et al. (2006, Algorithm 18) works as follows.
<sub>8</sub> Consider the projection $\mathrm{DPP}(\omega(x)\mathrm{d}x, K_N)$ as defined in our Section 2.2, with $K_N(x,y) = \Phi(x)^\intercal\Phi(y)$ where
<sub>9</sub> $\Phi(x) \triangleq (\phi_0(x), \dots, \phi_{N-1}(x))$. This DPP has exactly $N$ points, $\mu$-almost surely. To get a valid sample $\{X_1, \dots, X_N\}$,
<sub>10</sub> it is enough to apply the chain rule to the vector $(X_1, \dots, X_N)$ and forget about the order. The vector has density

$$\frac{\det[K_N(x_p, x_n)]_{p,n=1}^N}{N!} \prod_{n=1}^N \omega(x_n) \overset{(5)}{=} \frac{K_N(x_1, x_1)}{N}\omega(x_1) \prod_{n=2}^N \frac{K_N(x_n, x_n) - \mathbf{K}_{n-1}(x_n)^\intercal\mathbf{K}_{n-1}^{-1}\mathbf{K}_{n-1}(x_n)}{N - (n-1)}\omega(x_n),$$

<sub>11</sub> where the RHS is precisely the chain rule. The challenge is twofold. First, one must use an efficient way to sample
<sub>12</sub> exactly from the conditionals in the chain rule. Second, one must efficiently evaluate the kernel $K_N$ (6). In our
<sub>13</sub> submission, we followed BH and used rejection sampling to sample the conditionals, using always the same proposal
<sub>14</sub> distribution $\omega_{\mathrm{eq}}(x)\,\mathrm{d}x$ (A.12) and rejection bound (A.14). But, unlike BH, we computed $K_N(x,y)$ more efficiently by
<sub>15</sub> coupling the slicing feature of the Python language with the dedicated `scipy.special.eval_jacobi`. This carefully
<sub>16</sub> avoided redundant evaluations of orthogonal polynomials (OPs) in evaluating the multivariate kernel, which were a
<sub>17</sub> bottleneck. Since the submission, we further applied tricks familiar to MLers, and stored the feature vectors $\Phi(x_{1:n-1})$
<sub>18</sub> to exploit the Gram structure when computing $\mathbf{K}_{n-1}(x_n) = \Phi(x_{1:n-1})^\intercal\Phi(x_n)$. Besides, we found out that using
<sub>19</sub> the marginal $N^{-1}K_N(x,x)\omega(x)\,\mathrm{d}x$ as a rejection sampling proposal allowed us to reduce the number of (costly)
<sub>20</sub> evaluations of the quadratic form $\mathbf{K}_{n-1}(x_n)^\intercal\mathbf{K}_{n-1}^{-1}\mathbf{K}_{n-1}(x_n)$. This new proposal can be sampled without too many
<sub>21</sub> OP evaluations since it is a mixture, where each mixture component involves only one OP and can be sampled using
<sub>22</sub> rejection sampling again, this time using the original proposal $\omega_{\mathrm{eq}}(x)\,\mathrm{d}x$ of BH and the rejection bound (A.13). After
<sub>23</sub> making implementation improvements, getting one sample of a multivariate Jacobi ensemble with $N = 1000$ points in
<sub>24</sub> dimension $d = 2$ now takes less than a minute, compared to hours with the original implementation of BH (2016).

<sub>25</sub> All these improvements hold for all continuous DPPs, and together resulted in the dramatic speedups that we observed.
<sub>26</sub> A more specific improvement for OP-based kernels and $d = 1$, has been to implement Theorem 2 of Killip & Nenciu
<sub>27</sub> (2004), which surprisingly reduces DPP sampling to diagonalizing a simple tridiagonal random matrix. Finally, as noted
<sub>28</sub> by **R2**, the code we provided substantially helps; this code will be made public as a fully documented Python toolbox.

<sub>29</sub> **R1** *Theoretically, the authors shed light on the classical EZ method that utilizes DPP for numerical integration.*

<sub>30</sub> Indeed, one of our contributions is to formally link EZ and DPPs. This needed to be done because $(i)$ DPPs were
<sub>31</sub> formalized 15 years after EZ (1960), and $(ii)$ while the theory of DPPs has been thoroughly investigated in the 2000s,
<sub>32</sub> the work of EZ had been mostly forgotten by then, probably because sampling looked an insurmountable obstacle.
<sub>33</sub> All recent DPP-based numerical integration works were seemingly unaware of EZ. Now that the link is made and the
<sub>34</sub> methods are "aligned", as **R1** writes, we can imagine several lines for future work, like borrowing DPP machinery to
<sub>35</sub> prove a central limit theorem for EZ, or further connecting EZ to Bayesian quadrature.

<sub>36</sub> **R1** *the method still does not seem to be applicable to practical tasks with high dimension or large sample size.*

<sub>37</sub> As remarked by **R3**, DPP-based Monte Carlo algorithms are new and further work is certainly needed to make them
<sub>38</sub> practical. However, we believe that the EZ estimator combined with fast DPP samplers is already of practical interest
<sub>39</sub> for high-dimensional integration of functions that are known to be sparse in some basis of $L^2$.

<sub>40</sub> **R1** *I cannot tell whether (a) Theorem 1 has new results or is a known result listed for a modern proof (b) Section 3.1*
<sub>41</sub> *contains novel results (c) it is a novel idea to use orthonormal polynomials in EZ.*

<sub>42</sub> (a) Theorem 1 is indeed a known result and we bring a modern formulation and a modern proof. As discussed above, our
<sub>43</sub> reformulation highlights the unknown fact that EZ is actually based on sampling from a DPP. We provide an efficient
<sub>44</sub> exact sampling procedure for the multivariate Jacobi ensemble with ideas that benefit the general case (b) Section
<sub>45</sub> 3.1 only recalls known results from BH (2016), to ease the later comparison with EZ. Comparing nonasymptotic and
<sub>46</sub> asymptotic variances brings insight, for instance. (c) It's not novel: orthogonal polynomials are quite common in
<sub>47</sub> approximation theory, where the EZ method comes from.

<sub>48</sub> **R1** *I expect (a) one experimental result for Sections 4.2-3. [...] (b) comparisons with i.i.d. Monte Carlo or MCMC [...]*

<sub>49</sub> (a) Sections 4.2 and 4.3 indeed each contain an experiment. We designed these experiments to go beyond the toy
<sub>50</sub> illustration of BH (2016) and showcase the differences between EZ and BH. Extensive plots are to be found in Sections
<sub>51</sub> B2–6 of the appendix. (b) We will add the i.i.d. baseline, which will help visualize the faster rates of BH and EZ.

<sub>52</sub> **R1** *Is it possible to apply DPP to dynamics-based MCMC and particle-based variational inference methods?*

<sub>53</sub> In principle yes, DPPs have the potential to yield generic variance reduction in importance sampling. The kernel needs
<sub>54</sub> to be carefully chosen, though, if one wants faster-than-Monte-Carlo rates like with BH or EZ.

[Meta-Review · NeurIPS 2019]

This paper establishes links between determinantal point processes (DPPs) and Monte Carlo techniques (e.g., for application in Bayesian inference), developing new results and insights for DPP-based quadrature methods. The paper is written primarily from a theoretical perspective but also contains numerical experiments. A strength of the paper is that it links various ideas from the relevant literature together in a manner that a NeurIPS audience will appreciate. The reviewers identified some weaknesses in the paper, such as the need to more clearly establish how the results and ideas in the paper are related to prior work. The authors are strongly encouraged to incorporate these suggetsions from the reviewers when revising the paper for the final version.